# Liquid Metal Embrittlement Susceptibility of Hot Formed Zn-Al-Mg Coated Steel with Eutectic Coating Microstructure

Yubo Yang [1,2], Yu Fu [1,2], Guangxin Wu [1,2], Hongliang Liu [3], Yu Chen [3], Qun Luo [1,2,*] and Qian Li [1,2,4]

1   State Key Laboratory of Advanced Special Steel, School of Materials Science and Engineering, Shanghai University, Shanghai 200072, China; yuboyang@shu.edu.cn (Y.Y.); yuclaire@shu.edu.cn (Y.F.); gxwu@shu.edu.cn (G.W.); shuliqian@shu.edu.cn (Q.L.)
2   Shanghai Key Laboratory of Advanced Ferrometallurgy, School of Materials Science and Engineering, Shanghai University, Shanghai 200072, China
3   Technical Center of Ben Gang Group Corporation, Benxi 117000, China; daliang_lhl@hotmail.com (H.L.); chenyu830409@163.com (Y.C.)
4   National Engineering Research Center for Magnesium Alloy, Chongqing University, Chongqing 400044, China
*   Correspondence: qunluo@shu.edu.cn

**Abstract:** Liquid metal embrittlement (LME) in Zn-based coating plates during hot stamping is an abnormal phenomenon where intimate contact between liquid Zn and a steel matrix results in the penetration of liquid Zn into the matrix, causing ruptures. In order to alleviate LME phenomenon, this paper designed a series of eutectic Zn-Al-Mg coating alloys to improve the uniformity of the Zn element distribution in the coating during heat treatment and inhibit the reaction between Fe and Zn. The high temperature mechanical properties of the coated steels are determined using thermal simulation to calculate the relative reduction in fracture energy, which is used to evaluate the LME level of the different composition coatings. It is suggested that the Zn-4.5Al-3.0Mg coating shows the highest LME resistance at 920 °C. The microstructure of these Zn-Al-Mg coated steels is observed after austenitization at 850 °C~920 °C for 3 min, which shows that the uniformity of the microstructure after austenitizing is affected by the composition of the coating and the austenitizing temperature. The higher temperature benefits the homogenization of the coating and steel and inhibits LME. The findings of this study provide valuable insights for the development of ternary Zn based LME resistant coatings.

**Keywords:** liquid metal embrittlement; thermodynamic calculation; Zn-Al-Mg alloy; hot dipping

## 1. Introduction

Third generation advanced high strength steels were used in automotive manufacturing owing to their high ductility and fatigue resistance [1–11]. Good corrosion resistance [12] of sheet steel was achieved by galvanizing [6,8,13–16] compared to Al-Si coatings. However, the use of Zn-based coatings led to liquid metal embrittlement (LME). LME is an abnormal phenomenon in which the liquid metal penetrates into the solid metal after close contact of some specific solid-liquid metals, which leads to a great reduction in the plasticity and even to the fracturing of the solid metal [1,5,7,8,17–21]. The phenomenon usually occurs in the process of hot stamping [1,7,8,15,19,20,22,23], resistance spot welding [2–6,18,24], and other high temperature processing [25,26]. It can usually be concluded that three factors are required for the occurrence of LME: (1) aggressive metals with low melting points (relative to the matrix), such as Bi, Ga, and Zn, becoming a liquid at high temperatures during heat treatment; (2) a matrix in which liquid metal can easily penetrate; and (3) the presence of stress inside or outside the matrix (usually lower than the yield strength of the matrix) [25,27,28].

In an Fe/Zn system, the specific process for the formation of LME crack was that stress created an additional diffusion flux, which enhanced the liquid Zn penetration in addition

to the diffusion flux generated by the concentration gradient, and as the stress increased, the stress diffusion flux became more dominant [19]. In the presence of stress, Zn atoms penetrated into the grain boundary ahead of the progressing LME-crack in the steel plate, and subsequently [29], because that compared with the special grain boundaries, there were more Zn atoms and impurities segregated at the high-misorientation-angle random grain boundaries [30]. The Zn atoms occupied the Fe grain boundary sites and weakened the adjacent Fe-Fe atomic bonds [31], which decreased the grain boundary's fracture toughness [32], resulting in a fracture on one side of the Zn-doped grain boundary [31]. Then, through the continuous supply of liquid Zn, the $\Gamma_1$ or $\delta$-phase was formed during cooling, and the thermal stresses increased due to the large mismatches in the thermal expansion coefficients between the various intermetallic phases [33]; microcracks occurred at the interfaces between the intermetallic phases or between the intermetallic phases and the matrix [6].

In light of this, several attempts have been made to suppress LME by grain boundary engineering (GBE) [34]. The suppression of the intergranular degradation under two sets of low-strain-heat-treatment processing conditions (grain boundary engineering application conditions) is believed to eliminate the "weak" high-energy random GBs and instead induces the "stable" low-energy coincidence site lattice (CSL) GBs. Besides, changing the heating processing conditions, including the austenitizing temperature, heating rate, holding time, strain rate, stamping temperature and so on, was also recognized as a viable approach to improve resistance against LME susceptibility. With an increasing holding time at high temperatures or decreasing heating rate [1], more Fe-Zn intermetallic compounds with higher melting temperatures were formed, which made the substrate less exposed to liquid Zn; therefore, LME susceptibility was suppressed. However, it is worthy to note that most of the high temperature processing involves relatively extreme conditions, such as a high heating rate and austenitizing temperature, which implies that it is not effective to suppress LME by changing the conditions in an industrial environment. Furthermore, regarding the alloy composition of the coating, a method was proposed to adjust the Al content in the Zn-xAl-1.8Mg coating to increase the thickness of the Fe-Al inhibition layer, thereby retarding the interdiffusion of Fe and Zn, aiming to improve this key issue for an industrial application. However, research revealed that with an increasing heating temperature and prolonged heating time, the Fe-Al intermetallic compound layer may gradually break down, leading to the penetration of liquid Zn into the steel substrate, resulting in the occurrence of LME phenomenon [35]. In addition, it was found that the uniform $\alpha$-Fe(Zn, Al) layer formed during the high-temperature tensile process of Zn-Al-Mg coated plates can prevent the direct contact of the liquid metal with the steel substrate, thereby completely suppressing LME at high temperatures [20]. Therefore, it is necessary to develop LME-resistant Zn-based coatings, which can effectively inhibit the formation of LME cracks under extreme conditions.

In this work, CALPHAD-type calculations were performed to help in the design of the Zn-Al-Mg coating alloys. The solidification simulation was calculated based on the thermodynamic description of the Zn-Al-Mg system. Thermal stretching is used to evaluate the LME susceptibility of the different composition coating steels. The microstructure of the Zn-Al-Mg coatings at high temperature of 850 °C~920 °C were determined to analyze the LME resistance mechanism.

## 2. Experimental and Calculations

### 2.1. Design of Zn-Al-Mg Coating Alloys

In order to suppress the infiltration of liquid Zn into the steel substrate during hot stamping, an Al element was added to the Zn bath to preferentially form an Fe-Al intermetallic compound layer on the coating-substrate interface. A Mg element was added to the Zn bath to obtain a low potential phase $MgZn_2$, increasing the corrosion resistance of the coating. Simultaneously, the presence of a eutectic microstructure in the coating alloy

can prevent element segregation and further reduce the possibility of liquid Zn diffusion into the steel substrate.

Pandat software was used for the thermodynamic calculations of the Zn-Al-Mg ternary system based on the thermodynamic description of the Zn-Al-Mg system [36]. The liquidus projection was calculated for the preliminary design of the coating alloy, while the solidification paths were used to obtain the phase composition of the designed alloys after solidification.

The liquidus projection of the Zn-Al-Mg ternary system is shown in Figure 1a. It can be seen that the Zn-rich corner of Zn-Al-Mg ternary phase projection is divided into four primary phase regions, including HCP-Zn, FCC-Al, $Mg_2Zn_{11}$, and $MgZn_2$. Meanwhile, the eutectic curves between the two primary phases and ternary eutectic points are presented in it. Compared with the multiphase microstructures with coarse primary phases, the eutectic microstructure would be more homogeneous, which avoided the concentration of Zn element in some primary phases or intermetallics. Therefore, the coating alloy compositions were designed along the eutectic curves. Furthermore, the melting point of the designed coating alloys was limited to below 420 °C in order to minimize the optimization of the hot-dipping processing parameters. Finally, four kinds of alloys were chosen as marked in Figure 1a.

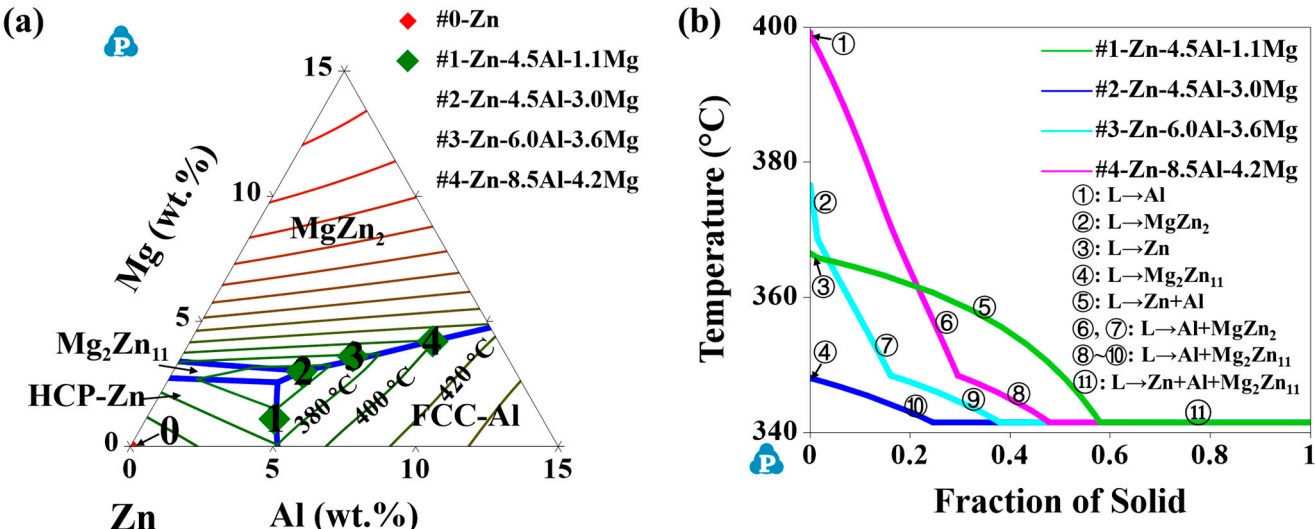

**Figure 1.** (**a**) Zn-rich corner of Zn-Al-Mg ternary phase projection, (**b**) solidification path and phases of alloys screened.

The solidification paths of the designed coating alloys as shown in Figure 1b were calculated to obtain the content of the eutectic microstructure. It was calculated using the Scheil model because of the rapid cooling rate. From the beginning to the end of the solidification, four alloys underwent multiple-phase transformations and finally solidified at 341 °C with a ternary eutectic reaction. According to the solidification results, the main microstructures were still the eutectic for the four alloys. The specific content of the eutectic in alloys is listed in Table 1. #1-Zn-4.5Al-1.1Mg alloy contains 56 mol% HCP-Zn + FCC-Al binary eutectic and 42 mol% HCP-Zn + FCC-Al + $Mg_2Zn_{11}$ ternary eutectic microstructure. Then, #3 and #4 contains three kinds of eutectic microstructures, FCC-Al + $MgZn_2$, FCC-Al + $Mg_2Zn_{11}$, and HCP-Zn + FCC-Al + $Mg_2Zn_{11}$, while #2-Zn-4.5Al-3.0Mg alloy contains 75 mol% ternary eutectic microstructure and 25 mol% FCC-Al + $MgZn_2$. Among those alloys, #2 alloy had the most content of the ternary eutectic microstructure.

**Table 1.** The content of eutectic in alloys (mol%).

| Samples \ Structure | HCP-Zn/FCC-Al | FCC-Al/MgZn$_2$ | FCC-Al/Mg$_2$Zn$_{11}$ | HCP-Zn/FCC-Al/Mg$_2$Zn$_{11}$ |
|---|---|---|---|---|
| #1 | 56 | \ | \ | 42 |
| #2 | \ | \ | 25 | 75 |
| #3 | \ | 15 | 22 | 62 |
| #4 | \ | 30 | 18 | 52 |

### 2.2. Materials and Test Methods

The investigated high-strength steel was a 1.2 mm thick PHS2000 steel plate produced by Ben Gang Group Corporation, Benxi 117000, China with a mechanical strength of 2 GPa. The official chemical composition of it is shown in Table 2. The hot-dipping was performed to attach a Zn-Al-Mg coating onto the surface of the steel plate. The coating alloys, located at the eutectic curves and eutectic point according to the phase diagram, were prepared using pure Zn (99.95 wt.%), Al (99.99 wt.%), and Al-50Mg intermediate alloy. All the alloys were melted in a crucible-type resistance furnace, followed by natural cooling. The actual composition of the alloys was detected using an Inductively Coupled Plasma Emission Spectrometer (ICP, 7300DV). Subsequently, these alloys were subjected to a heating and melting process in preparation for hot-dipping. Before hot-dipping, grinding, alkaline washing, pickling, and plating auxiliary was carried out to remove any surface contaminants from the steel sheets. The equipment employed for hot-dipping was a crucible-type resistance furnace. The applied temperature for hot-dipping, which corresponded to the heating and melting temperature of the coating alloys was 100 °C higher than the melting point temperature of the Zn-Al-Mg coating alloys. The hot-dipping time was 10 s. A Zn-Al-Mg coating was then dripped onto the surface of the steel plate.

**Table 2.** Chemical composition of PHS2000 steel (wt.%).

| C | Si | Mn | P | S | Al | Ti | V | N | B |
|---|---|---|---|---|---|---|---|---|---|
| 0.33 | 0.10 | 1.45 | ≤0.020 | ≤0.004 | 0.080 | 0.020 | 0.18 | ≤0.0040 | 0.0020 |

The high temperature tensile testing was completed using a Gleeble 3500 thermomechanical simulator machine to determine the thermomechanical state. The thermocouple was welded at the center of the sample to measure the temperature variation and a 15 mm extensometer was used to measure the strain. The heating treatment is illustrated in Figure 2a, which consisted of heating to the target temperature at a rate of 50 °C/s, isothermal holding for 3 min to ensure the complete austenitization of the steel plate, and then deformed in tension until failure using a stain rate of 0.5 s$^{-1}$. After fracturing, the sample was quenched to room temperature. Figure 2b shows the geometry and dimensions of the specimens for hot tensile testing.

The heat treatment testing of the Zn-Al-Mg coated plates was performed in a tubular furnace under air atmosphere. The austenitization temperature was set in the range of 850 °C to 920 °C. The samples were heated to the target temperature and held for 3 min, followed by water quenching to room temperature.

The cross metallographic analysis of the samples was prepared through mechanical grinding and polishing with a diamond suspension. The microstructure characterization of the Zn-Al-Mg coated steels, the coated steel plate after heat treatment, and the coated steel plate after high temperature tensile testing was observed using a Zeiss Sigma 500 field emission scanning electron microscope (SEM) operated at 20 keV and an OXFORD INCA energy dispersive X-ray spectroscopy (EDS).

The X-ray diffraction (XRD) was used to characterize the phase constituents of the coatings. The testing parameters were as follows: Cu target Kα radiation, voltage of 40 kV, current of 200 mA, scanning angle range from 10 to 90°, and scanning speed of 4°/min.

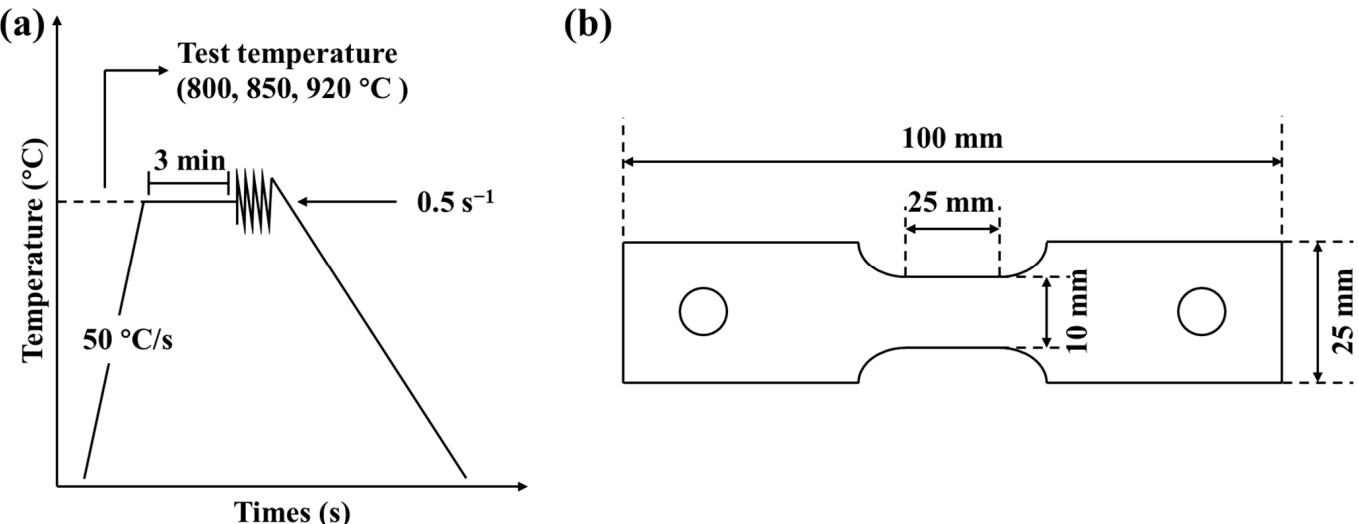

**Figure 2.** (**a**) Schematic representation of thermomechanical cycles used for hot tension tests of specimens, (**b**) schematic illustration of the hot tensile specimen geometry.

## 3. Results and Discussion

### 3.1. Microstructural Analysis of Zn-Al-Mg Coated Steel

The SEM micrographs along with the EDS analysis of the #1-Zn-4.5Al-1.1Mg coating are shown in Figure 3a. The #1 coating was observed to be a layer consisting of binary and ternary eutectic zones. The content of the binary eutectic is around 60%, which agrees with the results calculated in the solidification path. The ternary eutectic was distributed between the binary eutectic with a proportion of about 40%. Combining the solidification simulation, the EDS data, and the XRD test results, the binary eutectic was proved to be made up of η-Zn and α-Al (binary eutectic: η-Zn + α-Al), while the ternary eutectic was composed of η-Zn, α-Al, and an intermetallic compound $MgZn_2$ (ternary eutectic: η-Zn + α-Al + $MgZn_2$), which does not include the $Mg_2Zn_{11}$ as designed in the coating alloys. This is due to the relatively small $Mg_2Zn_{11}$ phase region in the Zn-Al-Mg ternary phase diagram, as shown in Figure 1a, making it more likely to form the larger $MgZn_2$ phase region during the actual solidification. The composition of each phase presented in Figure 3a is shown in Table 3. And the characterization results of the phases in the coating are presented in Figure 4. The EDS mapping was carried out as shown in Figure 3a to observe the distribution of the elements throughout the coating. The Zn and Mg elements were uniformly distributed across the thickness of the coating as shown in the images. The Al element, on the other hand, was locally enriched across the thickness of the coating, and the position of enrichment in the map scanning coincided with the position of the Zn/Al two-phase eutectic, which is consistent with the actual test results.

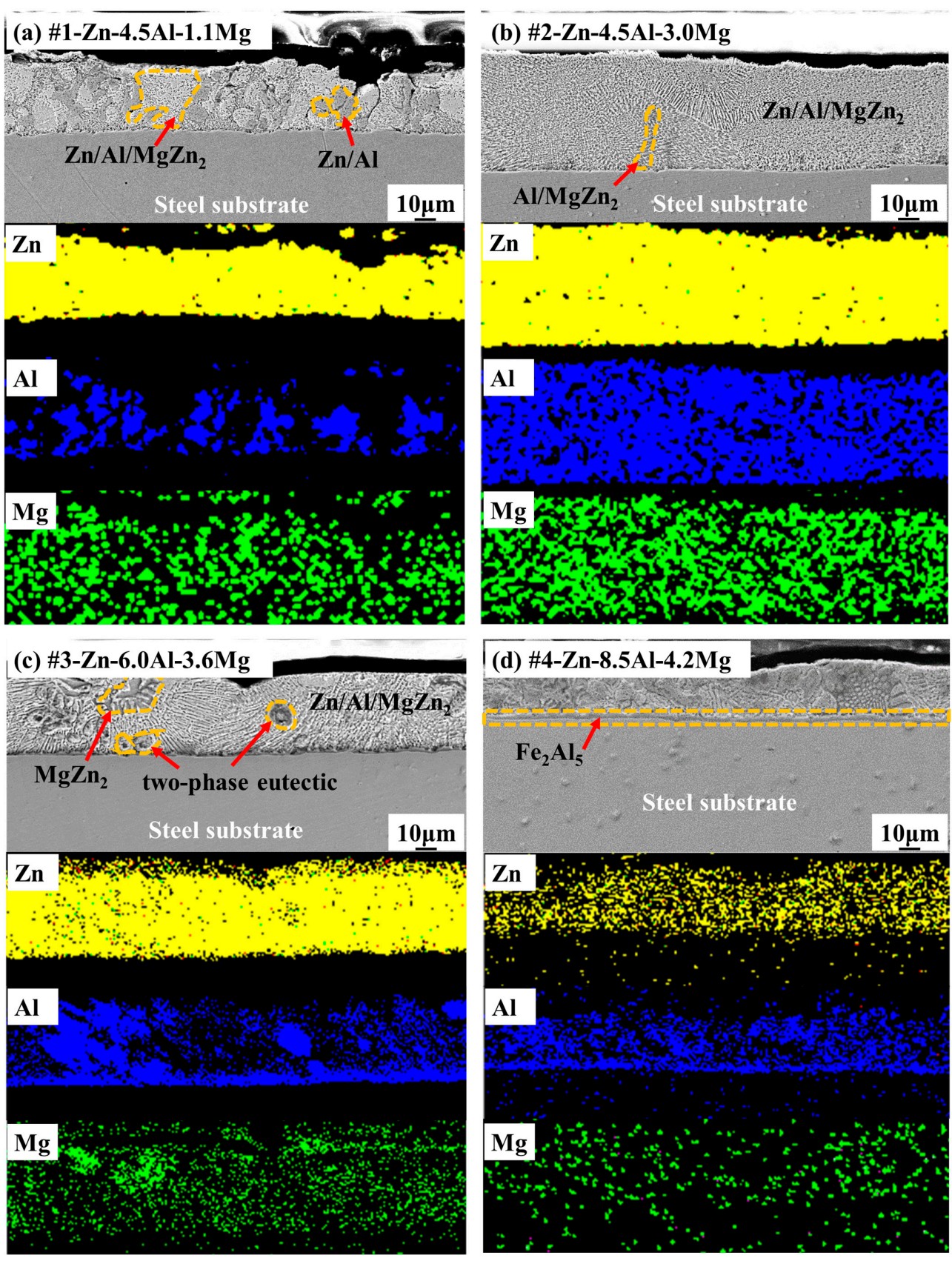

**Figure 3.** SEM micrographs and EDS analysis of (**a**) #1-Zn-4.5Al-1.1Mg, (**b**) #2-Zn-4.5Al-3.0Mg, (**c**) #3-Zn-6.0Al-3.6Mg and (**d**) #4-Zn-8.5Al-4.2Mg coatings.

**Table 3.** Composition of phases present within #1~#4 coatings obtained using EDS analysis (at.%).

| Coating | Region | Element | | | |
|---|---|---|---|---|---|
| | | **Zn** | **Al** | **Mg** | **Fe** |
| #1-Zn-4.5Al-1.1Mg | Binary eutectic | 55.1 | 33.9 | 0.4 | 0.3 |
| | Ternary eutectic | 88.6 | 3.3 | 0 | 1.0 |
| #2-Zn-4.5Al-3.0Mg | Three-phase eutectic | 75.9 | 13.8 | 3.7 | 0.5 |
| | Single phase | 53.8 | 3.7 | 29.7 | 0.3 |
| #3-Zn-6.0Al-3.6Mg | Two-phase eutectic | 28.4 | 55.1 | 0.9 | 0.2 |
| | Three-phase eutectic | 73.0 | 10.8 | 10.0 | 0.4 |
| | Two-phase eutectic | 36.0 | 57.1 | 0 | 0.9 |
| #4-Zn-8.5Al-4.2Mg | Three-phase eutectic | 67.5 | 7.0 | 4.3 | 1.0 |
| | Intermetallic compound | 4.6 | 67.9 | 0 | 23.1 |

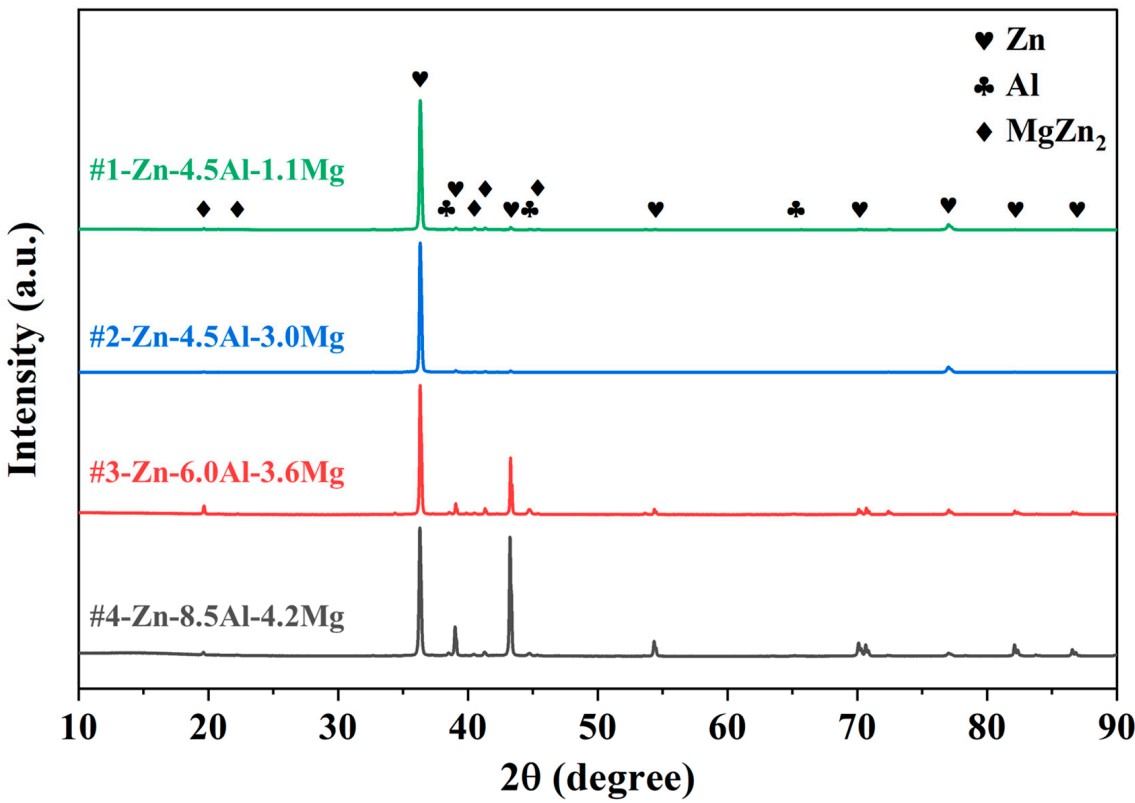

**Figure 4.** XRD patterns of #1-Zn-4.5Al-1.1Mg, #2-Zn-4.5Al-3.0Mg, #3-Zn-6.0Al-3.6Mg and #4-Zn-8.5Al-4.2Mg coatings.

The microstructure of the #2-Zn-4.5Al-3.0Mg coating is shown in Figure 3b. As can be seen in the micrograph, the coating displayed different morphological features to that of the #1 coating, with the microstructure size of the #2 coating being finer, and the three-phase eutectic was almost throughout the entire coating. The two-phase eutectic was surrounded by the three-phase eutectic, with a content of about 20%. The EDS maps showed a change in the distribution of the Al element compared to that of the #1 coating: the Al element was distributed in the coating homogeneously. Combining the EDS analysis results shown in Table 3, the XRD characterization results of the coating in Figure 4, and the solidification path simulation, it can be concluded that the three-phase eutectic consists of η-Zn, α-Al, and an intermetallic compound MgZn$_2$ (three-phase eutectic: η-Zn + α-Al + MgZn$_2$), α-Al and intermetallic compound MgZn$_2$ constitute two-phase eutectic (two-phase eutectic: α-Al + MgZn$_2$).

Figure 3c shows the SEM micrograph and the EDS analysis of the #3-Zn-6.0Al-3.6Mg coating, revealing the presence of different kinds of eutectic and intermetallic compounds at the interface. Compared with the #2 coating, the size of the microstructure in the #3 coating increased slightly owing to the decreased content of the three-phase eutectic. The total content of the α-Al + MgZn$_2$ eutectic increased to about 40%, and the η-Zn + α-Al + MgZn$_2$ three-phase eutectic only had a proportion of about 60%. Furthermore, the intermetallic layer formed by the Fe-Al elements at the interface was more evident.

Figure 3d shows the SEM micrographs of the #4-Zn-8.5Al-4.2Mg coating. The content of the three-phase eutectic observed in the coating is further reduced, the two-phase eutectic α-Al + MgZn$_2$, and the three-phase eutectic η-Zn + α-Al + MgZn$_2$ coexist in the coating with a ratio of 1:1, which leads to the further coarsening of the microstructure. Meanwhile, the thickness of the intermetallic compound layer increased, obviously with the increase in the Al content.

Based on the results of the phase diagram calculation and the morphology of the coatings, it can be seen that the coating with the highest content of a three-phase eutectic exhibits the finest morphology and the most homogeneous distribution of the elements. Moreover, the thickness of the Fe-Al compound layer at the interface increases gradually with the increase in the Al content in the coating.

### 3.2. The LME Behavior of Zn-Al-Mg Coated Steel

The influence of the coating composition on liquid zinc embrittlement was investigated at different temperatures. Figure 5a–c show the engineering stress-strain curves of bare and coated specimens obtained at 800 °C, 850 °C, and 920 °C with the stain rate of 0.5 s$^{-1}$. Figure 5d presents the evolution of the relative reduction in fracture energies ($(E_{bare} - E_{HG})/E_{bare} \times 100\%$) as a function of the testing temperature for the five coated steels used in this study. The relative reduction in fracture energy is a parameter for describing the severity of embrittlement; the fracture energy of each sample is $E = \int \sigma d\varepsilon$ [37]. $E_{bare}$ represents the fracture energy of the bare substrate, and $E_{HG}$ represents the fracture energy of the hot-dipping galvanized coated plate. The horizontal dashed lines represent the measurement scatter estimated to be around 10%, because the LME phenomenon would be pronounced if a relative reduction in the fracture exceeds 10% [37].

As shown in Figure 5a, it can be observed that the stress-strain curves obtained after tensile testing at 800 °C exhibit two types of tensile strengths. The tensile strength of all the coated specimens is approximately 30 MPa lower than that of the bare substrate. Additionally, they exhibit different elongations. The bare substrate has the highest elongation rate of 38%, while the elongation rate of the #0 coated specimen (pure Zn coated steel) is the lowest at 30%. The elongation rates of the #1~#4 coated specimens range between these two values, at approximately 34%. Therefore, as shown in Figure 5d, at 800 °C, the #0 coated specimen exhibits the lowest fracture energy value, with the highest relative energy reduction of 27%. This indicates that the #0 coated specimen has the highest LME sensitivity at this temperature. From this, it can be inferred that the eutectic structure formed by adding the Al and Mg elements in the #1~#4 coated specimens can suppress the occurrence of LME at 800 °C.

As the test temperature increases to 850 °C, the relative reduction in fracture energy values of the #1~#4 coated specimens significantly increase, indicating an increased LME sensitivity, as shown in Figure 5d, which is due to the increased difference in elongation rate between the coated specimens and the bare substrate compared to the 800 °C condition (the difference in elongation rates increases from 4% at 800 °C to 10% at 850 °C), as shown in Figure 5b. It indicates that 850 °C is the most severe condition for LME cracking, and the eutectic structure formed by adding the Al and Mg elements cannot effectively suppress its occurrence.

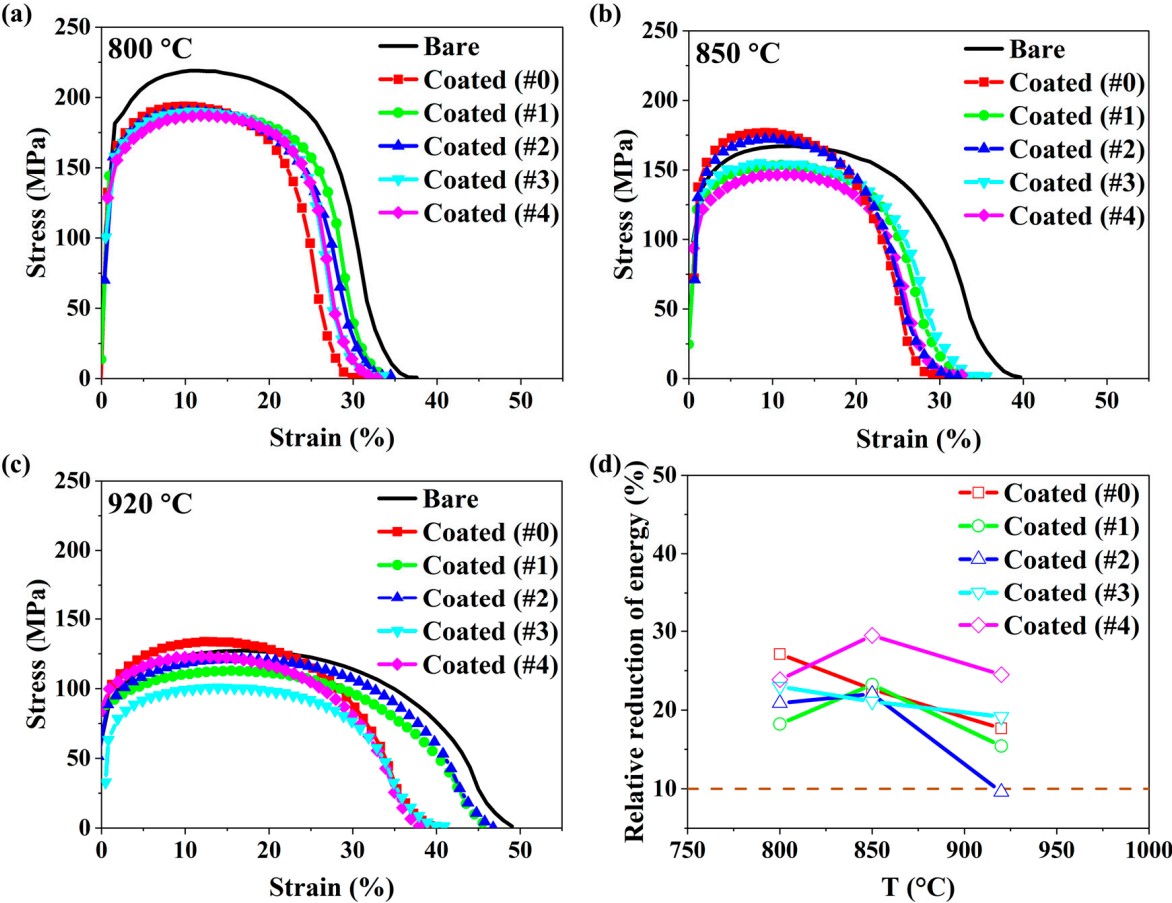

**Figure 5.** Engineering stress-strain curves of the #0-Zn, #1-Zn-4.5Al-1.1Mg, #2-Zn-4.5Al-3.0Mg, #3-Zn-6.0Al-3.6Mg, and #4-Zn-8.5Al-4.2Mg coated steels during HTTs at (**a**) 800 °C, (**b**) 850 °C, (**c**) 920 °C, and (**d**) influence on the embrittlement of the PHS2000 steel by liquid zinc.

The stress-strain curve after tensile testing at 920 °C, as shown in Figure 5c, demonstrates that the #2 sample exhibits mechanical behavior very similar to the bare substrate. Therefore, by calculating its relative energy reduction value, it can be observed that the value for the #2 coated specimen is below 10% of the horizontal dashed line, significantly lower than the values of the other coated specimens shown in Figure 5d. This indicates that the #2 coated specimen has the highest resistance to LME. It suggests that the coating with the highest content of ternary eutectic, the finest microstructure, and the most uniform element distribution can effectively suppress LME cracking. Furthermore, the relative energy reduction values of the #1, #3 and #4 coated plates under the tensile condition of 920 °C are lower than those at 800 °C and 850 °C, which indicates that 920 °C is a favorable condition for suppressing LME.

It can be observed that the LME sensitivity of the coating plate increases first and then decreases with the increase in test temperature near the austenitizing temperature. Additionally, the coated specimens with the finest microstructure and the most uniform element distribution exhibit the highest resistance to LME.

Figure 6a shows the SEM image of the fracture surface of the #2-Zn-4.5Al-3.0Mg coated specimen after HTT at 920 °C. The fracture surface exhibits a very thin edge, which shows a total ductile fracture. No sign of cracks was found on the coated surface, further confirming the ability of the #2 coated alloy to suppress LME cracking. As shown in Figure 6b, the cross-section SEM image of the fracture reveals a prominent necking phenomenon, also indicating the ductile fracture. Moreover, in the EDS image, no segregation of the Zn elements was observed in the interface of the coating and steel substrate.

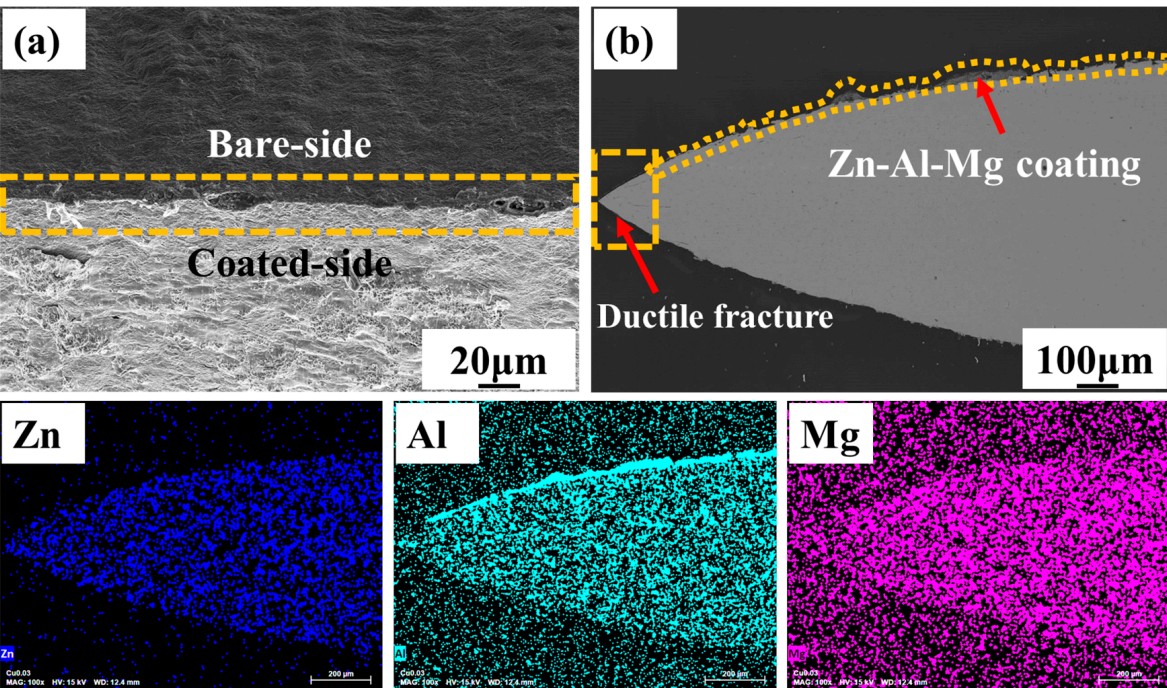

**Figure 6.** SEM micrographs of the (**a**) fracture surface and (**b**) cross-section of the #2-Zn-4.5Al-3.0Mg coated specimen after HTT at 920 °C.

### 3.3. Microstructure during High-Temperature Heat Treatment

The cross-sectional microstructure and EDS analysis of the #1-Zn-4.5Al-1.1Mg coated specimen after heat treatment at 850 °C for 3 min are shown in Figure 7a. It can be observed that the binary eutectic and the ternary eutectic structures present in the as-received coating have completely disappeared. The coating microstructure has coarsened entirely and different layers can be distinguished at the coating, including a surface layer with coarse equiaxed grains, an intermediate layer with dense equiaxed grains, and a layer with coarse columnar grains underneath. In addition, the coarse microstructure in the upper and lower layers contain numerous voids, resulting from rapid diffusion or evaporation of the liquid metal. It must be emphasized that, according to the EDS maps, the distribution of elements has also changed compared to the as-received samples. Zn elements tend to accumulate at the interface between the coating and the substrate, while Al elements exhibit enrichment on the surface of the coating. The Al elements segregate at the surface of the coating mainly due to the selective oxidation of Al. Furthermore, according to the EDS line scan results, the intensity of Al elements decreases while the intensity of Zn elements increases along the direction of the arrow in Figure 7a, which is consistent with the EDS surface scan results. When the heat treatment temperature is increased to 920 °C, the SEM micrograph shows a change in morphology of the coating compared to that at 850 °C as shown in Figure 7b. The coarse columnar structure at the bottom becomes finer and the coarse equiaxed structure on the surface has disappeared. It suggests a significant reduction in the coating thickness. More importantly, the elements Zn, Al, and Mg are uniformly distributed in the coating without any segregation phenomenon. Considering the LME sensitivity of the #1-Zn-4.5Al-1.1Mg coated specimen at 850 °C and 920 °C, it can be inferred that the uniform distribution of the elements in the coating after heat treatment can effectively suppresses LME cracking.

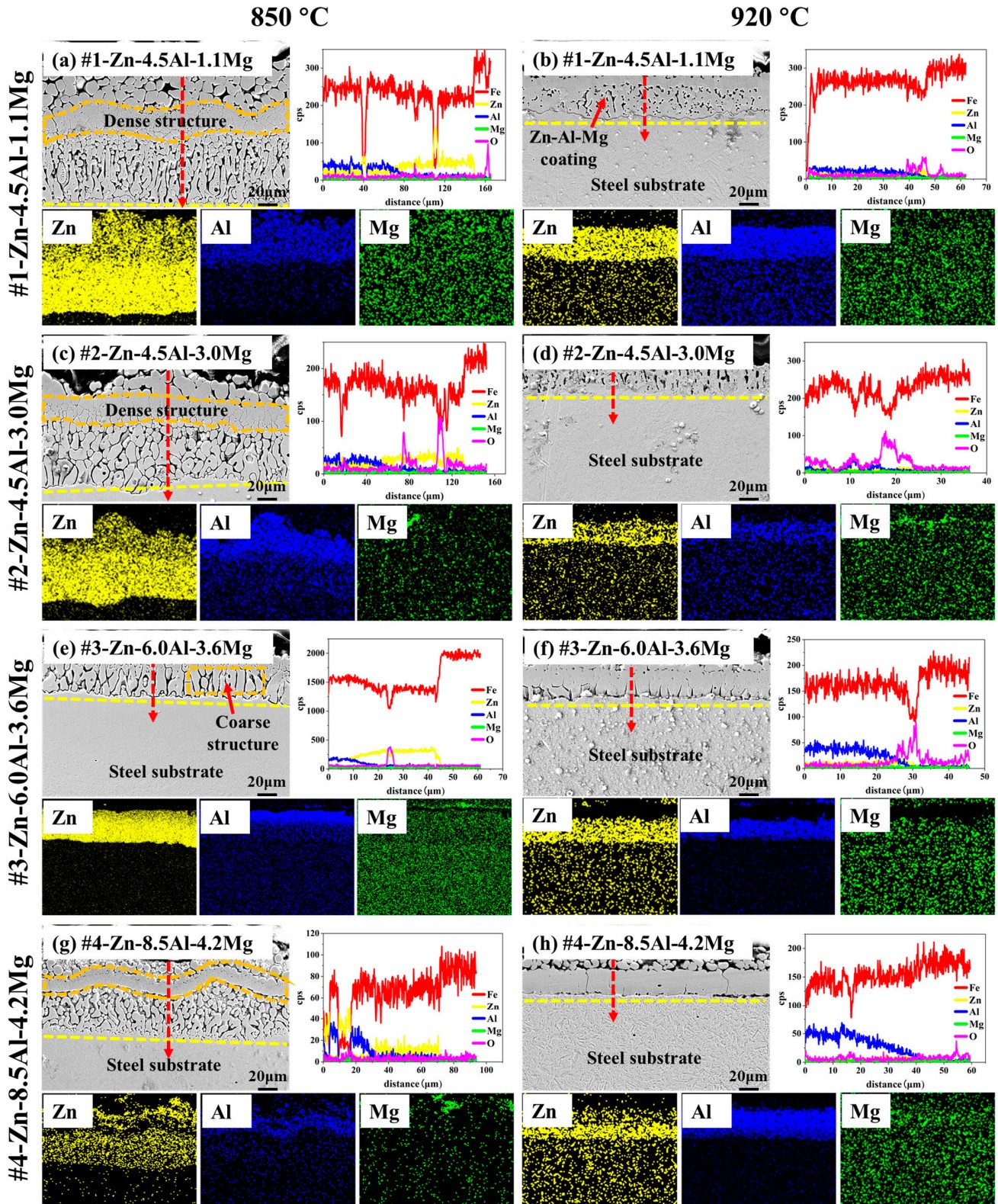

**Figure 7.** SEM micrographs and EDS analysis of (**a**,**b**) #1-Zn-4.5Al-1.1Mg, (**c**,**d**) #2-Zn-4.5Al-3.0Mg, (**e**,**f**) #3-Zn-6.0Al-3.6Mg, and (**g**,**h**) #4-Zn-8.5Al-4.2Mg coated PHS2000 steel after heat treatment at 850 °C and 920 °C.

Figure 7c shows the SEM image and EDS characterization results of the cross-section of the #2-Zn-4.5Al-3.0Mg coated specimen after heat treatment at 850 °C. The coating displayed similar morphological features to the #1 coated specimen after heat treatment at 850 °C but with a finer microstructure. The EDS analysis results indicate that Al elements also segregate towards the surface of the coating, while Zn elements segregate towards the interface. For the #2 coated specimen with the lowest LME sensitivity, after undergoing heat treatment at 920 °C, the microstructure of the coating completely transforms into a single-phase structure. Additionally, it shows a more uniform distribution of elements compared to the #1 coating, as shown in Figure 7d. The experimental results mentioned above provide evidence that the uniform distribution of elements after heat treatment can indeed effectively suppress the occurrence of LME cracking in the coated specimens.

As the content of Mg and Al elements increases in the coating, the microstructure of the #3-Zn-6.0Al-3.6Mg and #4-Zn-8.5Al-4.2Mg coatings becomes finer after heat treatment at 850 °C shown in Figure 7e,g. The Fe-Al intermetallic compounds at the interface of as-received samples disappeared and the coating thickness decreases compared to that of #1 and #2 coatings. Moreover, the Zn and Al elements are still segregated in the lower and the upper layers of the coating, respectively, which leads to higher LME sensitivity in the #3 and #4 coatings at 850 °C. After heat treatment at 920 °C, the SEM images of the #3 and #4 coatings reveal a dense single-phase microstructure. However, the elements Zn and Al still exhibit a tendency to segregate at the interface and the surface of the coating, respectively, similar to their distribution after the 850 °C heat treatment, which corresponds to the LME sensitivity observed at 920 °C.

Combining the above results with the microstructure of the untreated coating, it can be concluded that the heat treatment of the coatings with a fine eutectic microstructure and uniform elemental distribution leads to the formation of microstructures with uniform elemental distribution after high temperature annealing, suppressing LME cracking behavior.

## 4. Conclusions

The present study provides an analysis of the element distribution changes in the eutectic coatings we designed after heat treatment at different temperatures to investigate the LME cracking behavior of the coating. The designed coatings were susceptible to LME cracking at 850 °C, but the coatings offered resistance to LME cracking at 920 °C, especially the Zn-4.5Al-3.0Mg coating with a totally eutectic microstructure. The coating microstructure of the sample tested at 850 °C revealed the presence of different layers. The ductility in the sample decreased due to the segregation of the Zn element at the lower layer of the coating. The eutectic microstructure in the Zn-4.5Al-3.0Mg coating was completely dissolved and replaced with a uniform structure, which had a homogeneous distribution of the Zn element, successfully mitigating the LME problem by inhibiting the high reactivity of the liquid Zn with the steel substrate. As a result, no LME cracks were observed in the Zn-4.5Al-3.0Mg coated plate samples with the total eutectic microstructure, and the failure in the samples exhibited characteristics of ductile fracture. The significance of substantially reducing LME sensitivity in automotive steel at high temperatures is highlighted by the findings of this study, which focus on the importance of uniform element distribution in entirely eutectic Zn-based coatings with a specialized structure. By uncovering the reasons behind the remarkable resistance of eutectic Zn-Al-Mg coated steel to LME-induced cracking at elevated temperatures, this research provides innovative insights for the future design and production of LME-resistant Zn-based coatings.

**Author Contributions:** Conceptualization, Q.L. (Qun Luo) and Y.Y.; methodology, Y.Y. and G.W.; software, Q.L. (Qun Luo) and Y.Y.; validation, Y.Y., Y.C. and H.L.; formal analysis, Y.F.; investigation, Y.Y.; resources, Q.L. (Qian Li); data curation, Y.Y. and Q.L. (Qun Luo); writing—original draft preparation, Y.Y.; writing—review and editing, Y.Y., Q.L. (Qian Li) and Y.F.; visualization, Y.Y. and Q.L. (Qian Li); supervision, Q.L. (Qun Luo) and Q.L. (Qian Li); project administration, Q.L. (Qian Li) and Q.L. (Qun Luo) All authors have read and agreed to the published version of the manuscript.

**Funding:** This research was funded by the National Natural Science Foundation of China (U2102212 and U1908224), the Shanghai Rising-Star Program (21QA1403200), Liao Ning Revitalization Talents Program (XLYC2007066) and CITIC-CBMM Niobium Technology Extension Project (2022FWNB-30077).

**Data Availability Statement:** Available upon request.

**Acknowledgments:** The authors gratefully acknowledge School of Materials Science and Engineering of Shanghai University.

**Conflicts of Interest:** The authors declare no conflict of interest.

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
