# Peer review of "Liquid Metal Embrittlement Susceptibility of Hot Formed Zn-Al-Mg Coated Steel with Eutectic Coating Microstructure"

_metals, doi:10.3390/met13091523_

Round 1

Reviewer 1 Report

In this manuscript, different Zn-Al-Mg alloys were tested as a steel coating, studying the influence of type and fraction of eutectic, and temperature over the liquid metal embrittlement susceptibility. 

The authors exhibited a competent analysis but some points need improvements:

1- Please, check if references [11] and [26] are correct. In P. 1, line 29, it seems that the reference is not appropriate for the statement. The same in P. 1, line 36;

2- P. 1, line 37: Explain more about what are aggressive metals;

3- There is a lack of experimental details: about the Zn-Al-Mg alloys (chemical composition of the elements, the origin of the material, how they are made, etc.) and the dipping process (equipments, how it was made, etc.);

4- Why use the Zn-Al-Mg-Si system for the thermodynamic calculation? Are there no reliable studies on the Zn-Al-Mg system?;

5- In general, increase the size of the figures, it's difficult to see the details;

6- The EDS analysis didn't identify Mg at the ternary eutectic (η-Zn+α-Al+Mg2Zn11) but in the binary eutectic (η-Zn+α-Al) this element was detected. Why?;

7- Provide the EDS analysis (composition) of the other Zn-Al-Mg alloys;

8- How was the identification of phases with EDS? The phase characterization is weak, and other techniques should be used to confirm the intermetallics;

9- P. 6, line 208: Variables of the equation weren't explained;

10- What is the real range of temperatures in the experiments? The Abstract and  Experimental Procedures describe [850-950] °C but the Results and Discussion show [850-920] °C;

11- Why did the authors not investigate the microstructure in the condition of 900 °C? Why does LME sensitivity initially increase and then decrease?;

Reviewer 2 Report

Line 68-70: At this place, it should be added what compositions have already been tested by other researchers, because there is no justification why the Zn-Al-Mg composition was chosen for research described below.

Line 77: This chapter should be titled ”Materials and test methods”. To change!

Line 79: Chemical composition acc. to own analysis or acc. to the manufacturer’s catalog/certificate? Add this information in text or table. It is also worth presenting the mechanical properties of this steel.

Line 117: Chapter 3.1. should be moved to chapter 2, as it describes the justification for selecting the coating composition for testing. The results and analysis of the coating behavior (which is the goal of this study) begins in paragraph 3.2.

Line 322: In chapter “CONCLUSIONS” lacks conclusions. Be sure to add! Currently it is only a description of the results. There is no conclusion as to which composition is the best or in which direction further research should be conducted.

Reviewer 3 Report

In this research article, the authors have attempted to investigate the Liquid metal embrittlement susceptibility of hot formed Zn-Al-Mg coated steel with eutectic coating microstructure. The article needs minor revision and the comments are given hereunder

1. Please include a few lines/ literature review on coating technologies that can be used for preventing / mitigating Liquid metal embrittlement

2. Please include larger high resolution images of Figure 2.

3. XRD/ EPMA results must be included to elucidate compound formation.

4. Fracture toughness and hardness results should be included and discussed

In this research article, the authors have attempted to investigate the Liquid metal embrittlement susceptibility of hot formed Zn-Al-Mg coated steel with eutectic coating microstructure. The article needs minor revision and the comments are given hereunder

1. Please include a few lines/ literature review on coating technologies that can be used for preventing / mitigating Liquid metal embrittlement

2. Please include larger high resolution images of Figure 2.

3. XRD/ EPMA results must be included to elucidate compound formation.

4. Fracture toughness and hardness results should be included and discussed

Reviewer 4 Report

The topic is very actual. Liquid metal embrittlement (LME) of Zn-based coating plate during hot stamping  is discussed along with some calculations to design Zn-Al-Mg coating alloys. The paper is of sufficient originality due to the proposed methodology. The references are appropriate but should be updated since there are many recent papers in literature discussing this problem. In my opinion the authors  should describe  the aim more clearly as well as improving the conclusions.

Check english language

Round 2

Reviewer 1 Report

The authors answered all the points and considerably improved the manuscript.

Reviewer 3 Report

The revised version of the article can be accepted for publication

The revised version of the article can be accepted for publication